# Composite of Layered Double Hydroxide with Casein and Carboxymethylcellulose as a White Pigment for Food Application

**DOI:** 10.3390/foods11081120

**Published:** 2022-04-13

**Authors:** Estee Ngew, Wut Hmone Phue, Ziruo Liu, Saji George

**Affiliations:** Department of Food Science and Agricultural Chemistry, McGill University, Sainte-Anne-de-Bellevue, QC H9X 3V9, Canada; estee.ngew@mail.mcgill.ca (E.N.); wut.phue@mail.mcgill.ca (W.H.P.); ziruo.liu@mail.mcgill.ca (Z.L.)

**Keywords:** white pigment, E171 alternative, layered double hydroxides, casein, carboxymethyl cellulose, sustainability index

## Abstract

Titanium dioxide (TiO_2_) is commonly used in food, cosmetic, and pharmaceutical industries as a white pigment due to its extraordinary light scattering properties and high refractive index. However, as evidenced from recent reports, there are overriding concerns about the safety of nanoparticles of TiO_2_. As an alternative to TiO_2_, Mg-Al layered double hydroxide (LDH) and their composite containing casein and carboxymethyl cellulose (CMC) were synthesized using wet chemistry and compared with currently used materials (food grade TiO_2_ (E171), rice starch, and silicon dioxide (E551)) for its potential application as a white pigment. These particles were characterized for their size and shape (Transmission Electron Microscopy), crystallographic structure (X-Ray Diffraction), agglomeration behavior and surface charge (Dynamic Light Scattering), surface chemistry (Fourier Transform Infrared Spectroscopy), transmittance (UV–VIS spectroscopy), masking power, and cytotoxicity. Our results showed the formation of typical layered double hydroxide with flower-like morphology which was restructured into pseudo-spheres after casein intercalation. Transmittance measurement showed that LDH composites had better performance than pristine LDH, and the aqueous suspension was heat and pH resistant. While its masking power was not on a par with E171, the composite of LDH was superior to current alternatives such as rice starch and E551. Sustainability score obtained by MATLAB^®^ based comparison for price, safety, and performance showed that LDH composite was better than any of the compared materials, highlighting its potential as a white pigment for applications in food.

## 1. Introduction

E171 (food grade titanium dioxide (TiO_2_)) is one of the most widely used food additives containing nanoparticles [1]. Pigment grade TiO_2_ with its primary size ranging from 200–350 nm shows a high refractive index (2.6–2.9), negligible absorption in visible range of light spectrum, high scattering, and excellent masking power [2]. In addition, it is stable across a wide range of measurements of pH, temperature, and humidity, and it does not react with matrix components, making it a desired material for food and pharmaceutical products. E171 is copiously incorporated as a white pigment and a masking agent in over 900 commonly consumed food products such as dairy products and analogues, edible ices, confectionaries, surimi and similar products, food supplements, and seasonings and sauces [3,4].

TiO_2_ is generally recognized as safe (GRAS) according to the U.S. Food and Drug Administration (FDA) and is permitted to be used in food up to 1% without being declared on labels. Recent studies point towards potential adverse health outcomes from oral administration of TiO_2_ nanoparticles [2,5]. E171 particles are not designed to be nanoparticles, but pristine and recovered E171 particles from several food products showed a substantial proportion of nanoparticles (below 100 nm) [4,6]. The presence of nanoparticles in E171 has triggered concerns regarding the undesirable impacts of TiO_2_ nanoparticles on human health. In addition, recent publications from our group have reported the negative impacts of TiO_2_ nanoparticles on the functional integrity of intestinal epithelial cells and potential allergenicity of food allergens [7,8]. Pinget et al. also found that food grade TiO_2_ could impair the homeostasis in gut microbiota-host interactions [3]. The European food safety authority (EFSA) recently deemed that E171 is “not safe” as a food additive after evaluating the outcomes from recent health risk assessments [9]. Consequently, the European Union moved to ban the application of TiO_2_ as a food additive under Regulation (EC) No. 2022/63 in the Official Journal of the European Union, with a transition period until 7 August 2022. This and the growing consumer demand for natural and organic foods have resulted in food manufacturers using white pigment alternatives like rice starch (RS), silicon dioxide (E551), and calcium phosphate.

Layered double hydroxides (LDH) have recently fascinated researchers for their wide range of applications in various fields due to their biocompatibility, cost and resource effective methods of synthesis, and suitability for modification [10,11,12]. LDH is composed of positively charged two-dimensional (2D) metal hydroxide layers which stack alternatively with interlayers of anions, forming a three-dimensional (3D) “lasagna” structure. Structurally, the 2D “lasagna sheets” are composed of divalent cations octahedrally surrounded by OH–ions where the octahedra share edges forming an infinite 2D layer. The positively charged brucite-like (M^2+^ (OH)_2_) lasagna sheets are separated by an interlayer region containing charge compensating anions and solvation molecules, forming a 3D structure held together via electrostatic interactions and hydrogen bonds [13]. Anionic organic species like peptides, amino acids, and proteins could be intercalated in between the (M^2+^ (OH)_2_) layers through anion exchange chemistry [14]. LDH-composites thus generated have shown promising applications in medicine and energy conversion as well as storage and environmental remediation [15].

In this article, we report the generation of composite materials where Mg-Al LDH was intercalated with casein through anion exchange, and the surface was further modified with carboxymethylcellulose (CMC). LDH and its composites were characterized and compared to currently used white pigments for their physicochemical properties, masking power and safety. We report the superior performance, cost, and safety features of LDH composite in comparison to generally used E171 alternatives for its potential applications in products intended for human consumption.

## 2. Materials and Methodologies

Food additive silicon dioxide particles E551 (SIPERNAT 22) were obtained from Evonik Corporation (Parsippany, NJ, USA) and food additive titanium dioxide particles (E171, cat # 13463-67-7) were obtained from Minerals-Water, UK. All particles were used as received.

Commercial bovine milk casein (cat # 5890), magnesium oxide (MgO), aluminum oxide (Al_2_O_3_), sodium hydroxide (NaOH), rice starch (RS), and carboxymethyl cellulose (CMC) with 0.55–1.0 degree of substitution (DS) were purchased from Sigma Chemical Co. (St-Louis, MO, USA).

Stock solutions of particles (E171, E551) and rice starch were prepared by dispersing 10 mg/mL in deionized (DI) water obtained from a Milli-Q water system (Millipore Sigma, Milford, MA, USA).

### 2.1. Synthesis of Mg-Al LDH

Mg-based LDH was chosen because it is white color [16]; and Al^3+^ was chosen as the “guest” metal cation due to its ability to stabilize the α- form of hydroxides during LDH formation [17]. LDH was synthesized using a hydrothermal method as detailed elsewhere [18,19]. We chose the hydrothermal method as it improves the crystallinity of LDH which is desirable for the scattering of light within the LDH structure [20].

Accordingly, we mixed MgO: Al_2_O_3_ in a molar ratio of 2 in 50 mL of deionized water using ultrasonication for 30 m in a clean glass bottle. A Mg/Al ratio of 2 was chosen to form a clear hydrotalcite phase and higher basal spacing which is preferred for the anionic exchange reaction in the following steps [21,22]. One molar NaOH was added dropwise into the bottle under vigorous stirring until the pH of the solution reached ~10. The obtained mixture was stirred at 110 °C for 12 h in the tightly closed bottle, followed by ageing in the oven at 110 °C for 10 days for a dissociation-deposition-diffusion mechanism to mediate the formation of LDH [19,23]. The synthesized LDH was then washed with DI water twice at 12,500 rpm and the precipitate was freeze-dried overnight to obtain dry powder of LDH.

### 2.2. Synthesis of Mg-Al-Casein LDH (CLDH)

CLDH was synthesized by rehydration of LDH in the presence of casein. For this, 1.5 g LDH and 2.5 g casein were added to 100 mL DI water under vigorous stirring, followed by 30 min bath sonication to fully disperse the particles. NaOH (1 M) was added dropwise to the suspension under vigorous stirring. The suspension (pH~10) was further stirred in a tightly sealed bottle for 10 h. At this relatively high pH way above the isoelectric point of casein (pI = 4.6), casein was intercalated into the LDH interlayer region between the [Mg_2_Al (OH)_6_]^+^ “lasagna sheets”. Subsequently, the white suspension was collected by centrifugation at 12,500 rpm and washed with DI water twice to remove excess casein. The white precipitate from final wash was then freeze dried overnight to obtain CLDH.

### 2.3. Synthesis of Carboxymethyl Cellulose (CMC) Modified CLDH (CCLDH)

CMC (1.0 g) was dissolved in DI water (500 mL) with vigorous stirring at room temperature for 5 h for it to be fully dispersed. CLDH suspension in water (8 mg/mL) was added dropwise into the CMC solution such that the mass ratio of CMC to CLDH is 9 into a glass bottle. After 6 h, the CCLDH suspension was washed with DI water thrice, and was dried overnight in a freeze dryer.

### 2.4. Materials Characterization

X-Ray Diffraction (XRD) analysis was performed to observe and compare crystal structure of LDH, CLDH, and CCLDH composite and other particles used in this study. For this, the powdered particles were placed on the XRD specimen holder and pressed with a glass plate to fill the holder. The XRD patterns were recorded using a diffractometer (Bruker D8 Discover diffractometer with VANTEC–2000 detector system and Cu source) and analyzed with a diffraction angle range between 0 and 50° (2 theta). The voltage, current and pass time used were 40 Kv, 40 mA, and 1 s, respectively.

Attenuated total reflectance Fourier-transform infrared spectroscopy (ATR-FTIR, Bruker, Billerica, MA, USA) was used to identify the functional groups of the particles. For this, the dried powder samples were placed on the ATR probe and pressed. The FTIR spectra were taken in the range of 400–4000 cm^−1^, with a resolution of 4 cm^−1^ and 24 scans. Spectra were deconvoluted and analyzed using OMNIC software.

Transmission electron microscope (TEM) images were obtained using transmission electron microscope (Field Electron and Ion Company (FEI), Hillsboro, OR, USA) operating at 120 kV. For this, 1 mg/mL of particles (LDH, CLDH, CCLDH) were dispersed in ethanol and bath sonicated for 15 min. The suspension was further diluted in ethanol to 100 μg/mL. Five μL of samples were dropped on the TEM grid, air dried for 30 min, and imaged. Size distribution analysis for E171 was performed on at least 20 particles identified on the grid using ImageJ software (NIH, Bethesda, MD, USA).

The average hydrodynamic diameter, polydispersity index (PdI) and zeta potential of the particles were determined by dynamic light scattering (DLS) on a Zetasizer nano-ZS (Malvern Instruments, Malvern, UK) at 25 °C. The particles were dispersed and diluted to 50 μg/mL in deionized water and were measured twice. Size results are given as intensity distribution by the mean diameter with its standard deviation.

### 2.5. Transmittance and Opacity Measurements

The UV-Visible absorbance spectra could be used to associate with opacity [24]. The total reflectance spectra of the synthesized particles were measured and compared with E171, E551, and RS. For this, equal quantities of dry powders of particles were pressed onto the wall of the quartz cuvette of a UV–VIS diffuse reflectance spectrophotometer (Lambda 750, PerkinElmer Life and Analytical Sciences, Shelton, CT, USA) operated by UV WinLab™ software (Version 4.0, PerkinElmer Life and Analytical Sciences, Shelton, CT, USA), and % reflectance was measured in the range of 200–800 nm with 1 nm interval.

Opacity is defined as the proficiency of the pigment to mask the features of the underlying substrate [25]. Opacities of the synthesized particles were compared with that of E171 by measuring % transmittance. For transmittance measurement, 100 μL of particle suspensions (1 mg/mL) were dispensed into wells of 96 well plate and % transmittance at 550 nm was measured using a microplate reader (Spectra Max i3x, Molecular Devices, San Jose, CA, USA). Masking power comparisons between E171 and synthesized particles were conducted visually with optimized concentrations of hydroxy propyl methylcellulose (HPMC) (85% *w*/*v*) and plasticizer (Glycerol 10% *w/v*) for preparation of a film layer. Firstly, dry particles of E171, E551, and RS were suspended (1 mg/mL) in DI water, while the synthesized particles in dry powder form were suspended (1 mg/mL) at pH ~10 adjusted solution (with 0.1 M NaOH). These suspensions were individually mixed with slurry containing 85% hydroxy propyl methylcellulose (HPMC), 10% glycerol, and 5% solution. The resulting thick pastes (1 mL) of particles were spread evenly on a glass plate using a paint brush and left to dry overnight at room temperature. The glass plate was imaged against a worded background for visual comparison of masking power, recorded using digital camera.

### 2.6. Characterization of Stability of Particle Suspension to pH and Temperature

Stabilities of aqueous suspensions of LDH, CLDH, CCLDH, and E171, E551 (1 mg/mL) were evaluated as a function of time and under different pH and temperature conditions. For this, the particle suspensions were probe sonicated for three intervals of 15 s to evenly disperse the particles in DI water.

#### 2.6.1. Effect of pH

Stability of particle suspensions as a function of pH was performed by adjusting the pH of the suspension using 0.1 M NaOH and 1 M HNO_3_ to obtain pH levels of 3 and 9. These two pH levels were chosen as they represent the acidic and alkaline pH typically encountered in food processing. The suspension of particles with pH adjusted were left undisturbed for 30 min at room temperature, and the % transmittance (550 nm) was measured to assess the extent of particle sedimentation. The experiment was repeated in triplicates, and the mean value was plotted as % transmittance vs. pH.

#### 2.6.2. Effect of Temperature

Stability of particle suspension as a function of temperature was determined by heating 1 mg/mL samples in Eppendorf tubes in a thermostat water bath to 25, 50, 75, and 100 °C. After holding samples at the stipulated temperatures for 5 min, suspensions were cooled in ice bath to room temperature, and 100 μL of each suspension was aliquoted into a well of 96 wells plate. The % transmittance at 550 nm was determined using a microplate reader. The results of the three replicates were averaged and a graph of % transmittance vs. temperature was plotted with mean values and standard deviations.

### 2.7. Casein Antigenicity Study

One of the potential challenges for the application of the CCLDH composite would be the allergenicity of casein. Therefore, we compared the antigenicity of casein before and after conjugating with LDH using enzyme-linked immunosorbent assay (ELISA) as reported [8]. The ELISA plate was prepared by adding 100 µL of antigen solution (casein, CLDH and CCLDH (100 ng/mL) in coating buffer (0.1 M Na_2_HPO_4_ in DI water- pH 9.5)) to individual wells of a 96-well plate (Nunc Maxisorp 96-well ELISA plate). The plate was kept overnight at 4 °C for adsorption of antigen (casein, CLDH or CCLDH) to well surface. The uncoated antigen was discarded, and the residual free binding sites were blocked with blocking buffer (2% BSA in 1× TBS containing 50 mM Tris-HCl and 150 mM NaCl) for 1 h at 37 °C. After discarding the blocking buffer, primary antibody (1: 2000) (anti-casein rabbit antibody- cat # ab166596) was added in the blocking buffer and incubated for 1 h at room temperature. Excess of primary antibody (remaining in solution) was discarded, and the wells were washed four times with washing buffer (0.05% tween−20 in 1× TBS). The wells were then incubated with secondary antibody (1: 10,000) (anti-rabbit antibody- cat # 6721) in blocking buffer for 1 h at RT. Subsequently, the excess unbound secondary antibody was discarded, and the wells were washed four times with washing buffer. These wells were further added with 100 µL of substrate solution (5 mM of tetramethylbenzidine-TMB) and incubated for 3 min at RT for the enzymatic color development, followed by the addition of stop solution (100 μL of 2 M sulphuric acid) to terminate the reaction. Absorbance at a wavelength of 450 nm was determined by using a microplate reader. The amount of primary antibody binding onto antigen (casein) was determined using the standard curve, and the antigenicity is expressed as casein antigen response (ng/mL). The result presented is an average of values from triplicate experiments.

### 2.8. Cytotoxicity Study

Particle dispersion in serum-free cell culture media: Particle stock solution (1 mg/mL) was prepared by dispersing the particle powder in deionized water by probe sonication for 30 s of five intervals. The sonicated particle suspensions in DI water were further diluted in serum-free Dulbecco’s modified eagle medium (DMEM) medium.

Cell culture and co-incubation with particles: Human epithelial Caco-2 cells between passages 20–40 were cultured in DMEM (Invitrogen, Grand Island, NY, USA) supplemented with 10% (*v*/*v*) fetal bovine serum (FBS; Sigma Aldrich, St. Louis, MO, USA), and 1% antibiotics (penicillin/streptomycin; Invitrogen) 10 cm^2^ cell culture plates. These plates were incubated at 37 °C in a 5% CO_2_ incubator to reach 70–80% confluency. The cells were trypsinized, washed and seeded at 10,000 cells in 100 µL per well in a 96-well plate. Seeded cells were cultured overnight at 37 °C in a 5% CO_2_ incubator for complete attachment before particles exposure. The stock solution of particles dispersed in DI water was further diluted in serum-free DMEM medium, and particles exposures were carried out at concentrations of 25, 12.5, 6.25, 3.12, 1.56, and 0 µg/mL for 24 h.

Effect of particles on cell viability: Cytotoxicity of the particles on Caco-2 cells were assessed by the Alamar Blue (resazurin) assay after exposure to the desired concentration of the particles. For this, the exposure media was removed from the wells, and the cells were incubated with 100 µL resazurin solution (50 ppm) and incubated at 37 °C in 5% CO_2_ for 4 h. Absorbance at 570 nm was recorded using a microplate reader (Spectra Max M2, Molecular Devices, Sunnyvale, CA, USA). The readings were exported to an Excel file for comparison and statistical analysis.

Cytotoxicity was calculated as:(1)$$\mathrm{Cell}\;\mathrm{viability}\;\left(\%\right)=\frac{\mathrm{Abs}\;\mathrm{test}}{\mathrm{Abs}\;\mathrm{control}}\times 100$$

### 2.9. Application of Alternatives Assessment Framework

Developing sustainable alternatives to existing solution demands evaluation not only for performance but also for its cost and safety. The alternatives assessment framework allows us to identify sustainable alternatives to existing chemicals that needs replacement [26]. We conducted a preliminary alternative assessment of CCLDH for its potential application as a food pigment. The assessment conforms to the US National Academy’s Framework to Guide Selection of Chemical Alternatives. Accordingly, the Efficacy (% reflectance), Cost (USD/g) (Appendix A), and Safety (NOAEL) (Appendix A) were used for a multiparametric analysis using MATLAB, as detailed in Appendix A.

## 3. Results

### 3.1. Synthesis and Characterization of Particles

Figure 1a–f present the TEM images of particles tested in this study. In general, these particles had spherical or pseudospherical morphologies. Pristine LDH particles showed the characteristic flower-like morphology with hexagonal shape of the flaky sheets of LDH crystallites (Figure 1a). Significant changes in morphological features were observed when pristine LDH was modified by intercalation of casein. The presence of casein in the CLDH composite gave rise to the ordered inorganic lamellae represented by the darker lines (Figure 1b), which was similar to the Ca-Al-casein LDH reported earlier [27]. The schematic diagram (Figure 1g) shows the possible mechanism of casein intercalation through anion exchange when the negatively charged groups (–COO^−^) of casein gets electrostatically adsorbed between the positively charged LDH layers. The TEM micrograph of CCLDH (Figure 1c) showed mesoporous plates and rod-like cores of CLDH rendered by ordered CMC. Figure 1h demonstrates the scheme of CMC layered onto the CLDH particles. RS showed amorphous morphology with larger discrete particles with a size of ~5 μm (Figure 1d). E171 showed more consistent shapes with an estimated particle size of 100–120 nm (Figure 1e). Size distribution analysis (Appendix A) showed that ~20% of particles in E171 had size less than 100 nm (to be categorized as nanoparticles). Similar to previous observation from our group, E551 (food grade silica) were heavily agglomerated where the size of individual agglomerate was ~30 nm [28]. 

These particles were further characterized for their agglomeration behavior, surface charge, surface chemistry, and crystallinity. Based on the DLS results (Figure 2a), polydispersity of CLDH decreased significantly from pristine LDH but showed a significant increase after CMC adsorption. Since the LDH samples were poly-dispersed, it was hard to evaluate the mean size based on the TEM images; thus, the size distributions of the particles were measured in solution by DLS (Appendix A). The hydrodynamic diameter (Figure 2a) of pristine LDH was larger (>1 µm) in comparison to CLDH (<500 nm). The average diameter by volume (Figure 2a) of CCLDH had also increased compared to CLDH. The zeta potential (Figure 2b) of pristine LDH was +10.24 mV while that of casein intercalated (CLDH) was +6.03 mV. CCLDH showed a net negative zeta-potential of −11.96 mV. Zeta potential values of CLDH were close to those of pristine LDH than that of casein, suggesting that the LDH accommodated the casein molecules between the LDH brucite layers. CCLDH however showed a negative zeta potential value because of the surface adsorption of CMC onto CLDH.

ATR-FTIR was performed to assess the surface groups present in the LDH composites and the commercially used white food pigments, and the result is summarized in Figure 2c. The peak of 3692.09 cm^−1^ on the LDH graph features the vibration of −OH groups present in the LDH inorganic Mg (OH)_2_ layer [29]. The sharp peak at 1356.00 cm^−1^ indicates the adsorption of CO_2_ contamination from air during the collection process which is difficult to be excluded during LDH synthesis [30]. This contamination occupies some sites, which may lead to incomplete anionic exchange for casein intercalation [30].

The strong peak at 1388.11 cm^−1^ [due to ν(CO_3_^2−^)] and bands below 1000 cm^−1^ (due to M–O vibrations and M-O-H bending of LDH) are all characteristics of Mg-Al LDH [28,31,32,33]. The peak at 450.78 cm^−1^ in CLDH is a unique characteristic of Mg-Al LDH materials at 2:1 ratio [31]. This strong peak and a broad peak at 610.75 cm^−1^ are attributed to the lattice vibrations of Mg_2_Al-OH layer [34]. The band at 1443.18 cm^−1^ from CLDH is attributed to the stretching vibration of C = O symmetric stretching, verifying the presence of casein. The FT-IR spectrum of CCLDH is the combination of both the CMC and CLDH spectra, which is the good reason for the successful preparation of CCLDH. The casein peak of 1658.58 cm^−1^ had also broadened to 1617.08 cm^−1^.

The phase purity and crystallinity of the CLDH and CCLDH were analyzed and compared to the pristine LDH, alongside other industrial used white pigments by measuring XRD peaks (Figure 2d). In general, the LDH particles possessed defined basal diffraction peaks, suggesting their crystallinity. The varied phase composition of the synthesized LDH-composites indicated the presence of a variety of hydroxides and oxide hydroxides. The diffraction peaks at 15.3° for a rhombical symmetry is characteristic of hydrothermally synthesized LDH [19]. Strong diffraction peaks of the LDH due to a long aging time are in good agreement with those reported for Mg-Al-OH LDH [19]. The larger and more intense peak of LDH is attributed to its larger crystallite size compared to the smaller ones (less intense diffraction peaks), suggesting a correlation of size-crystallinity in LDHs [35]. Notably, the increased intensity of the rhombohedral symmetry peak at 23.6° in the composite material suggested successful intercalation of casein into the crystal structure. This finding concurs with the FTIR results as well as with the results from TEM images. LDH peaks disappeared in CCLDH suggesting the rendering of the negatively charged CMC over the positively charged surface of LDH [36]. All in all, physicochemical characterization of parental LDH and variants synthesized demonstrated a typical floral morphology of primary particle size from 100 to 1000 nm, well dispersed in aqueous solution where the original positive surface charge changed to negative when CLDH was surface functionalized with CMC, and the crystalline characteristics of the original LDH were reduced when it was intercalated with casein and subsequently functionalized with CMC. E551 particles were amorphous as evidenced by the broad peak in XRD spectra, while E171 were anatase [7]. RS, however, showed both crystalline and amorphous XRD patterns [36].

### 3.2. Reflectance, Stability of Particle Suspensions, and Masking Power

Among the synthesized particles, CCLDH showed excellent reflectance at the visible range which was second only to E171 (Figure 3a). This result complies with the masking power test whereby CCLDH had the strongest masking property in comparison with other LDH composites (Figure 4a). As shown in Figure 3a, the reflectance increased from 90% to 100% when the CLDH was engineered with CMC. 

Since food additives are commonly exposed to different temperatures and pH during processing or preparation, we investigated if the aqueous suspensions of particles tested were stable at elevated temperatures and in pH range [25,37]. We observed that suspensions of CLDH and CCLDH were more thermally stable compared to pristine LDH, possibly due to the free organic anion after casein intercalation. Further, adsorption of CMC onto the LDH surface could have prevented aggregation of LDH because of static and stearic hindrance. Both RS and E551 showed increased % transmittance as the temperature increased, which suggested that the opacity of currently used white pigment alternatives is not stable at elevated temperatures. In addition, the particle suspension seemed stable at alkaline and neutral pH, but an increase in % transmittance of the LDH composites was observed at pH 3 as LDH particles started to aggregate. Notably, RS was not stable at an alkaline pH as evidenced by an increased % transmittance. E551, however, was stable at an alkaline pH similar to that of the LDH composites.

The masking power of LDH composites was compared with E171 and other TiO_2_ alternatives by painting slurries containing particles on to a clean glass sheet and visually compared over a worded background. Even though the masking power was not as good as E171 (Figure 4a) LDH composites were superior to current alternatives used in the industry. Notably, pure LDH was not suspended well in the slurry and showed granular appearance while composites showed smooth smearing (Figure 4a). The aim of the research was to develop a white and opaque food pigment. Therefore, balancing the pigment structure and its stability in aqueous suspension is crucial. As seen in the time lapse images of particle suspension, casein intercalated LDH (CLDH) and modified with CMC (CCLDH) showed a more stable particle suspension.

### 3.3. Antigenicity and Cytotoxicity Assessment of LDH Composites

Since LDH particles are efficient delivery carriers of proteins, we also explored the possibility of using LDH as a modulator to prevent the binding of immunoglobulin E (IgE)-mediated hypersensitivity to casein which is often associated with milk allergy symptoms. As shown in Figure 5a, the antigen response of caseins intercalated in the CLDH reduced significantly which confirms its ability to suppress casein-specific IgE binding capacity, therefore decreasing the casein antigenicity. However the same response was not maintained in CCLDH, although the effect was still significantly lesser than pure casein.

The cytotoxicity response was evaluated based on the resazurin assay after the Caco-2 cells were treated with different concentations (0–25 μg/mL) of the synthesized pigments, its alternatives and E171 with cadmium chloride (CdCl_2_) as the positive control. The in vitro data clearly depicited that all of the synthesized LDH composites did not elicit any obvious cytotoxicity to Caco-2 with a viability above 80%. 

### 3.4. Comparison of Particles for Efficacy, Cost, and Safety

To demonstrate the applicability of the LDH composite as white pigments, we compared the tested materials in their performance, cost, and safety using MATLAB^®^ analysis (Figure 6). The reliability of E171 was scrutinized against its current alternatives and CCLDH based on three factors: efficacy, cost, and safety. Each factor was scored from 0 to 3 as detailed in Appendix A. The scores were used to develop a three-dimensional matrix using MATLAB^®^ (Figure 6). As seen in the heatmap, CCLDH was better in comparison to other white pigments in this multi-parametric comparison. E551 with its poor performance and cost had the lowest score, followed by E171 and rice starch.

## 4. Discussion

TiO_2_ is used copiously in several food, cosmetic, and pharmaceutical products as a white pigment. Recent health risk assessment studies, however, have indicated potential toxicity of dietary TiO_2_. For instance, oral administration of TiO_2_ nanoparticles was reported to cause epithelial hyperplasia and preneoplastic lesions in rodent models [5,38,39]. The obscurity on the health risk associated with TiO_2_ has incited the European Commission to request the European Food Safety Authority (EFSA) to reassess the safety intake of E171 (food grade TiO_2_) as a food additive. Thus, there is a surging demand for TiO_2_ (E171) alternatives in the food industry. Finding an alternative to TiO_2_ which has been extensively and widely applied across sectors is challenging from the perspective of meeting the constraints of performance, cost, and safety. Here, we report the synthesis and characterization of an LDH-based composite material as a potential alternative to E171 as a white food pigment.

LDH is a class of anionic clay particles with the characteristic hexagonal shape of flake-like sheets as revealed in the TEM image (Figure 1a). The flaky sheets are characteristic of the brucite-like layers, while irregular hexagonal shapes are due to more OH groups being exposed to the aqueous phase during ageing [40]. Mg-Al LDH was chosen as the constituent material due to its versatile properties such as high chemical stability, controllable particle size, and most importantly, it can be white in color [16]. While Balcomb et al., 2015, reported Zn-Al to be white, our results (data not included) showed that it was not as white as Mg-Al LDH after casein intercalation. Moreover, presence of Zn could compromise the safety of the material [41]. Casein, one of the major milk proteins when in micelle form scatters light and imparts opacity, contributing to the characteristic white color of milk. Therefore, we modified the pristine LDH by intercalating it with casein. Casein, which becomes anionic above its isoelectric point, is intercalated between the highly charged LDH layers through anion exchange (see scheme in Figure 1g). The homogeneous distribution and stabilization of the organic casein at the molecular level are driven by an integrated hydrogen-bonding network and Van der Waals interaction, as well as the host-guest interactions [42]. In particular, the negatively charged groups of casein (-COO^−^) would electrostatically interact with the LDH host layers, while its positively charged -NH^3+^ groups would repel, exposing the aqueous phase [43]. Comparing Figure 1a with b and inferring from the results of DLS, FTIR, and XRD, it is evident that the LDH lamellar structure acted as lasagna sheets having casein as the lasagna filling. The smaller agglomeration size of CLDH as compared to LDH (DLS data Figure 2a) and XRD data (Figure 2d) suggested that casein intercalation facilitated the dispersion of CLDH by changing the crystallographic structure of LDH. While the resulting CLDH showed improved suspension stability as depicted in Figure 4b, it was not on a par with E171. However, surface modification of CLDH with CMC prolonged the suspension stability of composite material suggesting the role of CMC in improving the electrostatic and steric hindrance-based repulsion between particles.

Light scattering at the boundaries within the 3D crystals of LDH is thought to play an important role in the white color appearance of the LDH. CLDH, however, has several types of interactions including stacking and charge interactions from sandwiching of casein molecules between the brucite layer, imitating a micelle-like structure. Surprisingly, CLDH exhibited a strong anti-reflectance behavior in the near UV region. This could be due to the incomplete anionic-exchange and increased amount of casein on the LDH surface, leading to a lower opacity. As described in the introduction, the “filling” of the interlayer lasagna sheets is mainly composed of water molecules and anions. When casein intercalation takes place, the former anion would be desorbed slowly from the LDH lattice. As the interlayer gets fully occupied, the excess casein could be exposed at the particle-water (aqueous phase) interphase [44]. Surface modification of CLDH with CMC (CCLDH) increased the reflectance possibly due to the ordered CMC acting as milk fat replacers entrapping the CLDH aggregates, forming a micelle-like structure [45]. The negative zeta potential of CCLDH could be due to the adsorption of CMC to CLDH through H-bonding between the hydroxide of LDH brucite layers and amide groups of the CMC molecules. In addition, the LDH composites suspensions showed high stability at elevated temperatures and alkaline pH after intercalation of casein due to the host-guest interaction involving hydrogen bonds as demonstrated by FTIR analysis. Addition of CMC also effectively protected the LDH particle suspension from heat-induced aggregation possibly because of steric stabilization mediated by multilayer structures [46]. Notably, LDH and its composites showed a general tendency of decreasing suspension stability as a function of decreasing pH. The basal surface is positively charged due to the isomorphous substitution of Mg by Al while the charges from the edges arise from the pH-dependent -OH groups. Therefore, a topotactic reaction may take place causing layer erosion at acidic pH [47]. This results in a decrease of electrostatic repulsions leading to the collapse or aggregation of LDH. In addition, the intercalated casein could be released as the LDH layers get delaminated at low pH [48].

Interestingly, intercalating the pristine LDH with casein improved its stability when dispersed in water, but this finding also evokes recognition of the LDH composite by the immune system. Based on the TEM micrographs in Figure 1a, the mixed (horizontal and vertical) platelet morphology allows the composite to form porous structure, acting as a protective layer encapsulating the casein molecule, thus attenuating the allergenicity of casein to a great extent [15]. Loading of casein into the interlayer galleries of the LDH octahedral sheets significantly reduced the antigenic IgE response to casein, therefore inhibiting the immunoreactivity. In comparison to CLDH, the antigenicity of CCLDH was not that apparent in comparison to casein. More studies are required to rule out the possibility of non-specific interaction between IgE and CCLDH containing CMC on the surface.

The predisposing factor of the successful application of an E171 alternative would be its efficacy in terms of intended function (expressed as % reflectance). However, industrial adaptation of the alternative food pigment should also consider other factors like scalability, reproducibility, cost, and safety. CCLDH, an anionic clay composite, is indeed an ideal white pigment alternative as shown in the alternative assessment analysis when these factors were considered (Figure 6). While this comparison was made on general grounds, more studies are warranted for determining the suitability of LDH composites in specific food applications.

## 5. Conclusions

Our findings contribute to the potential application of LDH composites as an alternative white food pigment. Mg-Al LDH was synthesized using the hydrothermal method. Masking power and suspension stability of the pristine LDH were substantially improved when casein was intercalated, followed by surface coating with CMC to form CCLDH. Casein antigenicity was also found to be suppressed when intercalated between the brucite layers. The ever-increasing demand for TiO_2_ alternatives as white pigment for food and pharmaceutical applications occurs under the constrains of today’s consumer choices on products that are not only delicious and visually appealing but also safe and better for human and environmental health. In this regard, results from the current study suggest promising applications of LDH composite as a white pigment for food, pharmaceutical, and cosmetic products. Nonetheless, the compatibility of LDH composites with common food ingredients and elaborate safety assessment using animal models remain to be explored.

## Figures and Tables

**Figure 1 foods-11-01120-f001:**
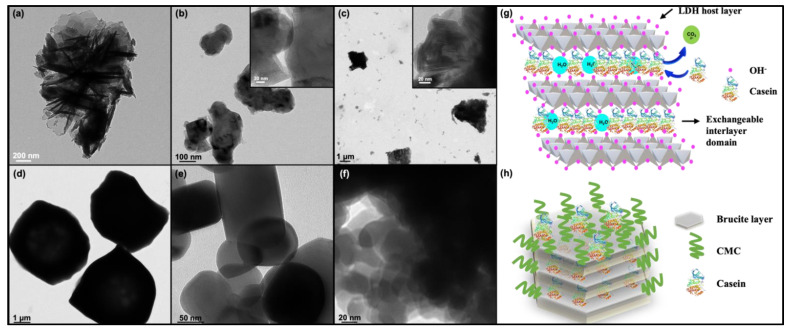
TEM micrographs of layered double hydroxide (LDH) powders: (**a**) LDH, (**b**) CLDH, (**c**) CCLDH, (**d**) Rice Starch, (**e**) E171, (**f**) E551, (**g**) scheme showing anionic exchange facilitating the intercalation of casein molecules between the brucite layers, (**h**) schematic of the CCLDH composite.

**Figure 2 foods-11-01120-f002:**
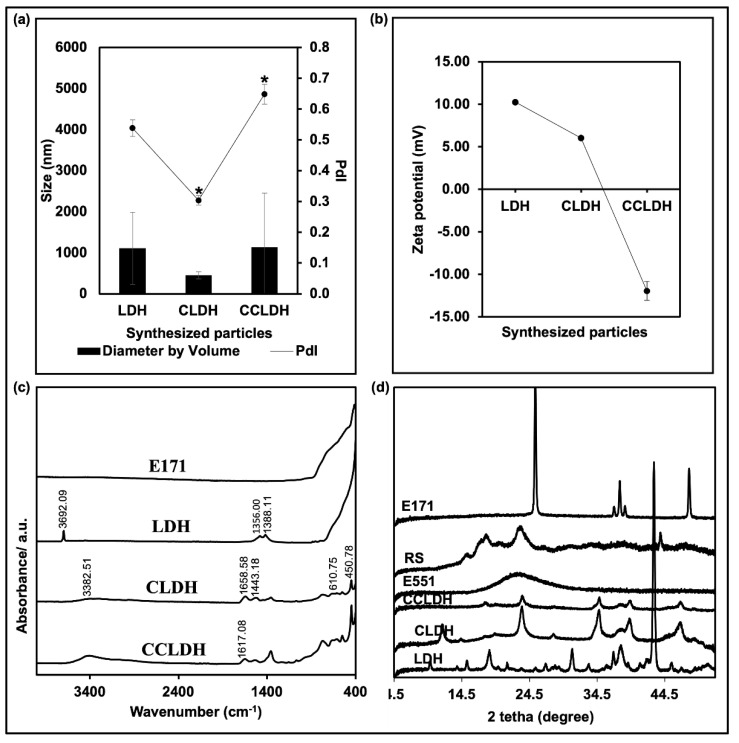
(**a**) Hydrodynamic diameter (bar graph) and polydispersity index (line graph) of particles (50 μg/mL) suspended in water. * Indicates statistical significance (*t*-test) in polydispersity (PdI) of CLDH in comparison to pristine LDH and CCLDH compared to CLDH, *p* ≤ 0.05, *N* = 2; (**b**) Zeta potential of LDH particles and its composites; (**c**) FTIR spectra of the particles; particles were placed directly on the ATR probe prior recording the spectra for a wavenumber range of 4000–400 cm^−1^ using ATR–FTIR; (**d**) XRD spectra of the tested particles. The spectra of dry powder of samples were captured using Bruker D8 Discovery X-ray Diffractometer.

**Figure 3 foods-11-01120-f003:**
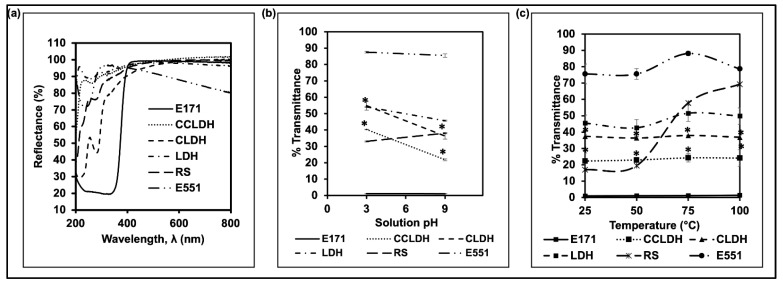
(**a**) Reflectance of powdered particles (both synthesized and commercially used white pigment) were measured in the range of 200–800 nm with 1 nm interval, demonstrating that all the synthesized LDH particles had a good reflectance in the visible range; (**b**,**c**) % transmittance (550 nm) at varied temperature and pH, respectively, suggested relatively high stability of LDHs at high temperatures, but not in acidic pH. * Indicate statistical significance (*t*-test) in % transmittance of LDH composites in comparison to pristine LDH, *p* ≤ 0.05, *N* = 3.

**Figure 4 foods-11-01120-f004:**
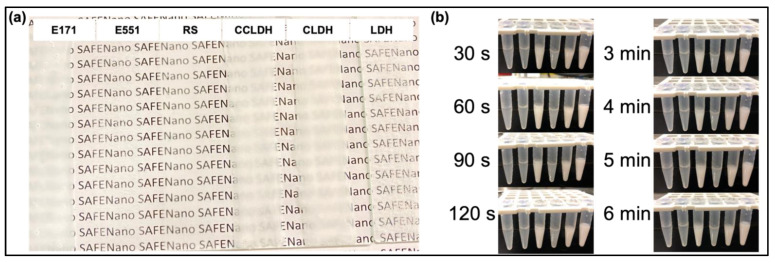
(**a**) Masking ability of the synthesized LDH and its composites was compared with that of commercially used E171 and its alternatives by suspending them into a mixture to form a thick paste to be painted on a glass plate and visually compared against a worded background. From left: E171, E551, RS, CCLDH, CLDH, and LDH. The LDH composites showed better masking. (**b**) Suspension stability of particles (from left: LDH, CLDH (1 mg/mL), CLDH (10 mg/mL), CCLDH (1 mg/mL), CCLDH CLDH (10 mg/mL), and E171) dispersed in DI water were visually compared from time-lapse images taken at 30 s intervals up to 180 s, followed by 60 s intervals from 3 min to 6 min.

**Figure 5 foods-11-01120-f005:**
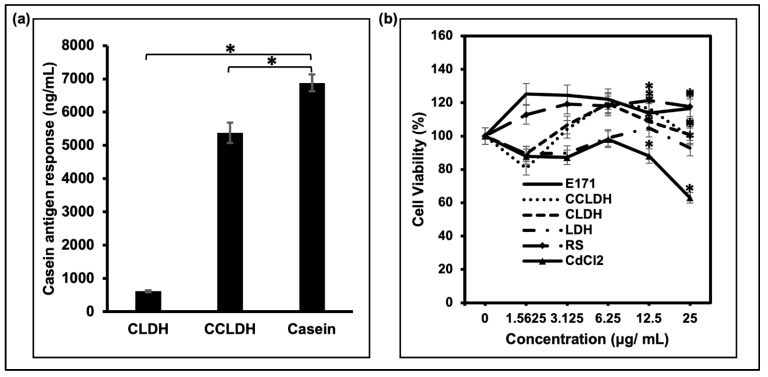
(**a**) IgE binding capacity was determined by ELISA, and its response was expressed as the casein antigen response (ng/mL) of casein samples under different LDH compounds. * Indicates statistically significant (*t*-test) difference in casein antigenic response of LDH composites with casein in comparison to raw untreated casein molecules, *p* ≤ 0.05, *N* = 3. (**b**) Particle concentration dependent Caco-2 cell viability changes post exposure to particles tested. The result suggested low/no cytotoxicity of LDH particles. * Indicates the significant differeces of the tested particles from the control (CdCl_2_) as verified by statistical *t*-test (*p* ≤ 0.05, *N* = 3).

**Figure 6 foods-11-01120-f006:**
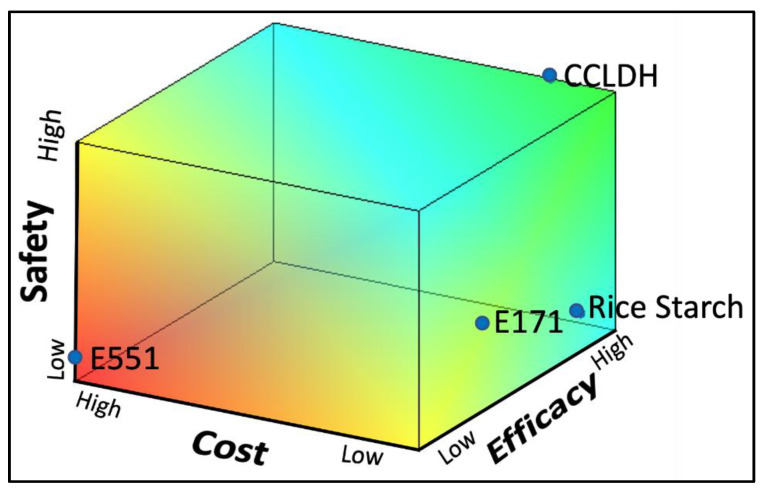
A three-dimensional matrix was developed by MATLAB^®^ for comparing the efficacy (% reflectance), cost (USD/g), and safety (NOAEL) of CCLDH to TiO_2_ alternatives and E171. The application desirability of the four particles in food are expressed in the phase diagram developed by MATLAB^®^, with red representing likely low success, blue likely high success, and green having considerable success as white food pigments. Position of each particle in the matrix was determined from individual scores for the three parameters assessed.

## Data Availability

Not applicable.

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
