# Peer review of "Composite of Layered Double Hydroxide with Casein and Carboxymethylcellulose as a White Pigment for Food Application"

_foods, 2022, doi:10.3390/foods11081120_

Round 1

Reviewer 1 Report

This manuscript is concerns the production of novel pigment particles. The mansuscript is well organized & suitable to a broad audience. The work is novel and the referencing should be commended for being up to date but more specifically not limited to only food science papers.

I do note that the materials and methods is quite short, and there isn't great depth into the why steps work and what are the limits around the preparation. e.g. at what pH range would I expect casein to coat the LDH particles. I assume this specific question would come down to the IEP and the authors elude to the electrostatics. Or if too much casein is adsorbed will that prevent the adsorption of CMC. I'm not particularly concerned about this point as this is probably a conseqeunce of writing an article with broad goals (everything from zeta potentials to antigen testing is covered).

Perhaps two comments I'd like to make is in relation to casein scattering.  Casein micelles will scatter light but individual casein proteins will not scatter. If individual caseins proteins are being adsorbed any change in opacity is likely a consequence of size increase. Maybe the authors could clarify if they have casein micelles or sodium caseinate.  Second in regards the charge. The adsorption of CMC and casein appears to electrostatically driven. Charge reversal should occur at sufficient coverage and if the adsorbed charge density is higher than that of the underlying particle. I don't have a reference I can suggest but I would suggest looking at the field polyelectrolyte adsorption on oppositely charged particles.  Helmuth Mowald (no relation to the reviewer) would have studied this phenomena 15-20 years ago.

All in all, you'll see these comments are very minor.

Reviewer 2 Report

This manuscript studies an E171 alternative which addressed an important need in the food industry. some moderate revisions are needed to address the following concerns: 

  1. the size distribution analysis is missing. it was reported that E171 has ~10-30% nanoparticles, however it was not shown in this study, mainly due to the smaller number of particle analyzed. all particles should have a size distribution analysis based on significant number of particles measured.
  2. another important concern is the E171sample. it is well known that the E171 samples are varied from sources to sources in terms of size and size distribution. How do you confirm the E171sample used in this study are a good representative sample?
  3. What is the expected shelf-life of this product? and compare to others?
  4. cost analysis of the CCLDH is not clear. please provide more details.
  5. more discussion is need regarding the pristine LDH with casein improved its stability in low pH. 
  6. compatibility study with common food ingredients may be a good future study. 

Round 2

Reviewer 2 Report

The authors provided additional data and addressed the comments well

Author Response

The authors have made substantial changes on spell checks and English language in the revised manuscript. Modifications can be seen on the "track changes" function on MS word.